# Bone and Soft-Tissue Sarcoma: A New Target for Telomerase-Specific Oncolytic Virotherapy

**DOI:** 10.3390/cancers12020478

**Published:** 2020-02-18

**Authors:** Hiroshi Tazawa, Joe Hasei, Shuya Yano, Shunsuke Kagawa, Toshifumi Ozaki, Toshiyoshi Fujiwara

**Affiliations:** 1Center for Innovative Clinical Medicine, Okayama University Hospital, Okayama 700-8558, Japan; 2Department of Gastroenterological Surgery, Okayama University Graduate School of Medicine, Dentistry and Pharmaceutical Sciences, Okayama 700-8558, Japan; shuyayano@okayama-u.ac.jp (S.Y.); skagawa@md.okayama-u.ac.jp (S.K.); toshi_f@md.okayama-u.ac.jp (T.F.); 3Department of Orthopedic Surgery, Okayama University Graduate School of Medicine, Dentistry and Pharmaceutical Sciences, Okayama 700-8558, Japan; joe@md.okayama-u.ac.jp (J.H.); tozaki@md.okayama-u.ac.jp (T.O.); 4Minimally Invasive Therapy Center, Okayama University Hospital, Okayama 700-8558, Japan

**Keywords:** oncolytic adenovirus, hTERT, autophagy, GFP, p53

## Abstract

Adenovirus serotype 5 (Ad5) is widely and frequently used as a virus vector in cancer gene therapy and oncolytic virotherapy. Oncolytic virotherapy is a novel antitumor treatment for inducing lytic cell death in tumor cells without affecting normal cells. Based on the Ad5 genome, we have generated three types of telomerase-specific replication-competent oncolytic adenoviruses: OBP-301 (Telomelysin), green fluorescent protein (GFP)-expressing OBP-401 (TelomeScan), and tumor suppressor *p53*-armed OBP-702. These viruses drive the expression of the adenoviral *E1A* and *E1B* genes under the control of the *hTERT* (human telomerase reverse transcriptase-encoding gene) promoter, providing tumor-specific virus replication. This review focuses on the therapeutic potential of three *hTERT* promoter-driven oncolytic adenoviruses against bone and soft-tissue sarcoma cells with telomerase activity. OBP-301 induces the antitumor effect in monotherapy or combination therapy with chemotherapeutic drugs via induction of autophagy and apoptosis. OBP-401 enables visualization of sarcoma cells within normal tissues by serving as a tumor-specific labeling reagent for fluorescence-guided surgery via induction of GFP expression. OBP-702 exhibits a profound antitumor effect in OBP-301-resistant sarcoma cells via activation of the p53 signaling pathway. Taken together, telomerase-specific oncolytic adenoviruses are promising antitumor reagents that are expected to provide novel therapeutic options for the treatment of bone and soft-tissue sarcomas.

## 1. Introduction

Adenovirus is a non-enveloped virus with a double-stranded DNA genome. There are multiple adenovirus serotypes [1], among which adenovirus serotype 5 (Ad5) is well characterized and has been genetically engineered to transduce transgenes in cancer gene therapy [2,3]. Although replication-deficient Ad5 is widely and frequently used as a safe virus vector in cancer gene therapy, this virus often shows a low transduction rate for transgenes, presenting a major obstacle that must be overcome to improve the efficacy of cancer gene therapy. Therefore, replication-competent Ad5 recently has been used as an effective virus vector in cancer gene therapy [2,3] and oncolytic virotherapy [4,5].

Oncolytic virotherapy has emerged as a novel antitumor treatment for eliminating tumor cells without affecting normal cells [6,7]. There are two bioengineering approaches for inducing tumor-specific cell death using Ad5. One approach is a modification of virus internalization. Ad5 enters target cells via binding of the viral fiber knob to the coxsackie and adenovirus receptor (CAR) protein on the surface of the target cell. As both tumor and normal cells often express the CAR protein on their surfaces, tumor-specific surface antigens are attractive target molecules for the tumor-specific internalization of Ad5. Therefore, modification of the viral capsid protein to bind to a tumor-specific antigen is a useful method for inducing the tumor tropism of virus infection. There are genetic, chemical, or mechanical engineering methods to enhance the infection ability of an adenovirus [8]. In contrast, the other approach for inducing tumor-specific cell death using Ad5 is by modification of virus replication. Ad5 replicates in the infected cells by inducing the expression of viral genes. Given that *E1*-deleted adenovirus has no replication ability, the adenoviral *E1* gene is a critical factor for replication of Ad5. Therefore, replacement of the wild-type *E1* promoter with the promoters of cancer-related genes is a useful method for enhancing the tumor tropism of virus replication [9,10,11,12]. There are many kinds of tumor-specific promoters, including those for the genes encoding human telomerase reverse transcriptase (hTERT) [13,14,15,16,17], midkine [18,19], cyclooxygenase-2 [20], and survivin [21]; insertion of each of these promoters can selectively enhance the replication of adenovirus in tumor cells. As the *hTERT* gene is frequently upregulated in a variety of malignant tumor cells with telomerase activity [22], the *hTERT* promoter is one of the most promising promoters for the tumor-specific expression of the viral *E1* gene.

For the development of tumor-specific oncolytic virotherapy, we have generated three types of *hTERT* promoter-driven replication-competent oncolytic adenoviruses: OBP-301, OBP-401, and OBP-702 (Figure 1). These viruses drive the expression of the adenoviral *E1A* and *E1B* genes under the control of the *hTERT* promoter to facilitate tumor-specific virus replication. OBP-301 (Telomelysin) exhibits a broad spectrum of antitumor effects in malignant tumor cells with telomerase activity [17]. To assess the spatiotemporal biodistribution of OBP-301, we have developed an *hTERT* promoter-driven oncolytic adenovirus, OBP-401 (TelomeScan), which induces the expression of the green fluorescent protein (GFP) in tumor cells [23]. Moreover, to enhance the antitumor efficacy of OBP-301, we have developed an *hTERT* promoter-driven oncolytic adenovirus, OBP-702, which induces the expression of the tumor suppressor *p53* gene in tumor cells [24].

Bone and soft-tissue sarcomas are rare diseases with the characteristics of heterogeneity and diversity [25,26]. They account for less than 1% of all adult solid malignant cancers [26]. There are more than 100 different histological subtypes of sarcoma [26]. The rarity and heterogeneity of bone and soft-tissue sarcoma represent a limitation in the development of novel molecular targeting therapy [27]. Therefore, a common molecular target is needed to develop novel treatment options for bone and soft-tissue sarcomas. As telomerase activity [28] and CAR expression [29,30,31] are frequently upregulated in bone and soft-tissue sarcomas, *hTERT* promoter-driven oncolytic adenoviruses are expected to be an attractive antitumor reagent against bone and soft-tissue sarcomas. In this review, we focus on the therapeutic potential of *hTERT* promoter-driven oncolytic adenoviruses for the treatment of bone and soft-tissue sarcomas. Moreover, the future directions of *hTERT* promoter-driven oncolytic virotherapy are discussed, especially for the treatment of metastatic bone and soft-tissue sarcomas.

## 2. Telomerase-Positive Type and ALT Type in Bone and Soft-Tissue Sarcomas

Telomerase is an enzyme that adds the telomere, a region of repeated nucleotides (TTAGGG in vertebrates), to the ends of chromosomes. Telomerase consists of a ribonucleoprotein complex containing two subunits, a catalytic subunit (in human, hTERT, and telomerase-associated protein 1) and an RNA subunit (human telomerase RNA component). Telomerase activity is closely associated with hTERT expression in tumor cells [32,33]. In normal cells without telomerase activity, cell division induces cell cycle arrest and senescence-related cell death via shortening of the chromosomal telomere end (Figure 2). In contrast to the case in normal tissues, telomerase is frequently and commonly activated in bone and soft-tissue sarcoma tissues [28]. Telomerase-positive sarcoma cells exhibit unlimited cell proliferation even after cell division via telomere elongation induced by activation of telomerase (Figure 2). Telomerase-positive sarcoma cells are suitable targets for *hTERT* promoter-driven oncolytic virotherapy. In contrast, a small population of sarcoma cells maintains their telomeres in a telomerase-independent manner via alternative lengthening of telomeres (ALT) [34]. ALT-type sarcoma cells show unlimited cell proliferation after cell division via telomere elongation induced by activation of homologous recombination (Figure 2). ALT-type sarcoma cells may be less sensitive to *hTERT* promoter-driven oncolytic virotherapy due to a lack of telomerase activity.

## 3. CAR Expression in Bone and Soft-Tissue Sarcoma Cells

Ad5 enters target cells primarily via binding of the viral fiber knob to the CAR protein. The efficacy of Ad5-based oncolytic virotherapy mainly depends on the expression level of CAR protein on the target cells [35]. CAR is frequently expressed in a variety of cancer types [36,37,38,39], including bone and soft-tissue sarcomas [29,30,31].

The in vitro antitumor efficacy of the Ad5-based *hTERT*-driven oncolytic adenovirus OBP-301 was tested in 14 bone and soft-tissue sarcoma cell lines with different levels of CAR expression (Table 1; eight osteosarcoma cell lines (OST, U2OS, HOS, HuO9, MNNG/HOS, SaOS-2, NOS-2, and NOS-10), two chondrosarcoma cell lines (NDCS-1, OUMS-27), one clear cell sarcoma cell line (CCS), one malignant peripheral nerve sheath cell line (NMS-2), one synovial sarcoma cell line (SYO-1), and one malignant fibrous histiocytoma cell line (NMFH-1)) [40]. Of these 14 lines, only those that were CAR positive also were sensitive to OBP-301. CAR expression correlated significantly with sensitivity to OBP-301. Thus, CAR expression is a critical factor for the treatment of bone and soft-tissue sarcomas by Ad5-based oncolytic virotherapy.

CAR expression often is downregulated during tumor progression [41,42] or when tumors are under hypoxic microenvironment [43]. Given that CAR-negative tumor cells are less sensitive to Ad5-based oncolytic virotherapy, a novel therapeutic strategy to target CAR-negative tumor cells is needed. There are two approaches to improve the infectivity of oncolytic adenoviruses for CAR-negative tumor cells. One approach is to upregulate the expression of CAR protein in tumor cells by treatment with histone deacetylase inhibitors [44,45,46]. The other approach is to modify the viral fiber knob to permit binding to cell surface molecules, such as integrins and CD46, other than CAR [47]. Moreover, it would be critical to assess CAR expression in tumor cells before initiating the treatment of bone and soft-tissue sarcomas.

We developed a telomerase-specific replication-competent oncolytic adenovirus, OBP-301 (Telomelysin), in which the tumor-specific *hTERT* promoter regulates the expression of the adenoviral *E1A* and *E1B* genes [17] (Figure 1). After infection, OBP-301 enters and replicates in tumor cells with telomerase activity due to activation of the *hTERT* promoter (Figure 3). Excessive virus replication then results in the induction of autophagy, cell lysis, and subsequent virus spread to the surrounding tumor cells (Figure 3). Preclinical experiments demonstrated that OBP-301 exhibits an antitumor effect against a variety of human cancer cell lines with telomerase activity, including bone and soft-tissue sarcoma cells [40,48]. Recent accumulating evidence has shown that oncolytic adenoviruses induce autophagy in association with cell death rather than cell survival [49]. OBP-301-mediated cytopathic activity is significantly associated with autophagy induction [50]. Regarding the molecular mechanism underlying oncolytic adenovirus-mediated induction of autophagy, we demonstrated that OBP-301-induced adenoviral E1A protein downregulates the expression of autophagy-suppressive epidermal growth factor receptor via upregulation of E2F1 and microRNA-7 [50]. Klein et al. showed that the oncolytic adenovirus Delta-24-RGD induces the activation of c-Jun N-terminal kinase (JNK) and phosphorylation of Bcl-2, leading to the suppression of Bcl-2/Beclin 1 autophagy suppressor complexes and autophagy induction [51]. Wechman et al. demonstrated that the adenoviral E1A and E1B-19K proteins induce the activation of JNK, resulting in the replication and oncolysis of Ad5 [52]. As JNK activation is responsible for the autophagic cell death induced by various factors [53], JNK-mediated autophagy may be involved in the antitumor efficacy of OBP-301-based oncolytic virotherapy.

## 4. Combination Therapy with OBP-301 and Chemotherapeutic Reagents for Osteosarcoma

Multi-agent chemotherapy is an important option for the treatment of osteosarcoma. Refractoriness to chemotherapy is a major obstacle to improvements in the clinical outcome of patients with osteosarcoma. OBP-301 enhances the sensitivity of osteosarcoma cells to the chemotherapeutic reagents doxorubicin and cisplatin [54]. Regarding the molecular mechanism underlying OBP-301’s enhancement of chemosensitivity, adenoviral E1A protein upregulates the expression of E2F1 and microRNA-29, leading to suppression of the anti-apoptotic myeloid cell leukemia 1 protein [54,55]. Thus, the combination of OBP-301 and chemotherapy is a promising antitumor strategy for the treatment of osteosarcoma.

In vivo experiments have demonstrated that invasive osteosarcoma cells cause bone destruction in an orthotopic xenograft tumor model [56,57]. Intratumoral injection of OBP-301 yields antitumor effects in subcutaneous and orthotopic xenograft tumors with bone and soft-tissue sarcoma cells [40]. However, bone destruction caused by OBP-301-resistant osteosarcoma cells was not suppressed by OBP-301 treatment [58]. Zoledronic acid (ZOL) is a third-generation bisphosphonate that is used clinically for suppressing bone destruction in patients with metastatic bone tumors [59]. Combination therapy with OBP-301 and ZOL efficiently suppresses bone destruction caused by OBP-301-resistant osteosarcoma cells [55]. Moreover, Conry et al. showed that ZOL treatment improves the progression-free survival in patients with metastatic osteosarcoma [60]. Thus, the combination of OBP-301 and ZOL is a therapeutic option, especially for the treatment of osteosarcoma with bone destruction and metastatic potential.

## 5. Detection of Tumor Cells by the OBP-401-Based GFP Induction System

To assess the biodistribution of OBP-301 in tumor cells, we developed a GFP-expressing oncolytic adenovirus, OBP-401 (TelomeScan), that selectively visualizes a variety of tumor cells, including bone and soft-tissue sarcoma cells, as GFP-positive cells [23,61,62]. The OBP-401-based GFP induction system is a useful method for the detection of tumor cells in both ex vivo and in vivo settings (Figure 4).

Ex vivo experiments showed that the OBP-401-mediated GFP induction system detects a variety of tumor cells in clinical samples of blood [63,64,65,66,67,68,69,70,71,72,73] and in peritoneal washes [74,75,76] from cancer patients. Gastric cancer patients with GFP-positive peritoneal tumor cells show poor prognoses [74]. As OBP-401-mediated GFP induction depends on telomerase activity and CAR expression in tumor cells, this system is a useful method to detect CAR-positive sarcoma cells, but not to detect CAR-negative sarcomas or normal cells [61]. Moreover, when combined with flow cytometry and genetic analysis, isolated GFP-positive tumor cells are available for the analysis of genetic alterations [64]. Thus, the OBP-401-based imaging system is a novel diagnostic option for the assessment of CAR expression and genetic alterations in bone and soft-tissue sarcoma cells.

In vivo experiments using xenograft tumor mouse models demonstrated that intratumoral injection of OBP-401 induces GFP expression in primary rectal tumors and in metastatic tumors at regional lymph nodes [23,77,78]. Intravenous, intrapleural, or intraperitoneal injection of OBP-401 also induces GFP expression in metastatic tumor cells [78,79,80]. In contrast, GFP expression is not induced in normal cells after infection with OBP-401 [23,61]. Thus, the OBP-401-mediated GFP induction system is a useful tool for the detection of tumor cells within normal tissues.

## 6. Fluorescence-Guided Surgery Using the OBP-401-Based GFP Induction System

Fluorescence-guided surgery (FGS) is an area of intense current interest in cancer treatment [81,82,83,84]. FGS has been shown to yield benefits, so curative strategies need to be developed for use in suitable clinical settings. The OBP-401-mediated GFP induction system is a useful method for marking tumor cells within normal tissues, facilitating treatment using FGS (Figure 4). In vivo preclinical experiments have demonstrated the utility of OBP-401-based FGS for various types of solid tumors [79,85,86,87,88,89], including osteosarcoma [90] and soft-tissue sarcoma [91]. In particular, the OBP-401-based FGS significantly reduced local recurrence and lung metastasis in orthotopic xenograft tumor models with human osteosarcoma and fibrosarcoma cells compared to bright-light surgery [90,91]. Thus, OBP-401 is a tumor-specific fluorescent labeling reagent that is expected to support the therapeutic potential of FGS.

Curative surgical resection is often limited to non-invasive tumors because the boundaries of tumor tissues often are unclear in invasive tumors, leading to the risk of residual tumor presence and local recurrence. The precise resection of bone and soft-tissue sarcoma also is difficult due to the invasivity of this cancer with respect to normal tissues. Although GFP is one of the most attractive fluorescent proteins, low penetration by the excitatory blue laser represents a limitation when illuminating deep tumor tissues. Another marker, KillerRed protein, is one of the brightest red fluorescent proteins, and is excited by a green laser with higher tissue-penetration ability [92]. We have generated a KillerRed-expressing oncolytic adenovirus, TelomeKiller, that permits the selective visualization of a variety of tumor cells, including sarcoma cells [93,94]. TelomeKiller may be a useful reagent for illuminating deep tumor tissues in FGS.

## 7. Antitumor Effect of OBP-702 in Association with the p53 Signaling Pathway

The p53 tumor suppressor protein is a multifunctional transcription factor that regulates diverse cellular processes, such as cell cycle arrest, senescence, apoptosis, and autophagy [95]. The *p53* gene is frequently inactivated due to somatic mutation in a variety of cancer types [96], including bone and soft-tissue sarcomas. Patients with Li-Fraumeni syndrome, who carry a germline mutation in the *p53* gene, are predisposed to bone and soft-tissue sarcomas [97]. Additionally, genetically engineered mice with mutated *p53* genes have been shown to predominantly develop bone and soft-tissue sarcomas [98]. These evidences suggest that the *p53* gene plays a critical role in the development of bone and soft-tissue sarcomas. Therefore, the restoration of the *p53* gene is expected to be an attractive therapeutic strategy for inducing strong tumor suppression [3,99]. A *p53*-expressing replication-deficient adenovirus vector, Ad-p53 (Advexin), previously has been developed as a way to introduce the *p53* gene into tumor cells in both preclinical and clinical settings [3]. However, the low transduction rate of the *p53* gene is a limitation in Ad-p53-based cancer gene therapy.

In human cancer cells, combination therapy with OBP-301 and Ad-p53 yields more profound antitumor efficacy than does monotherapy with OBP-301 [100], suggesting that the activation of the p53 signaling pathway enhances the antitumor efficacy of OBP-301. To prove this hypothesis, we have developed a *p53*-expressing oncolytic adenovirus, OBP-702, that strongly induces the p53 signaling pathway [24]. OBP-702 induces more profound antitumor effects in association with autophagy and apoptosis in osteosarcoma cells than does OBP-301 or Ad-p53 [58]. Regarding the molecular mechanism underlying OBP-702-induced autophagy and apoptosis, the p53-inducible modulator of autophagy (DRAM) and the pro-apoptotic BCL2 associated X apoptosis regulator (BAX) proteins are upregulated, but the p53-downstream target *p21* and *MDM2* genes are not upregulated in OBP-702-infected osteosarcoma cells (Figure 5). E1A-mediated upregulation of microRNA-93 and microRNA-106b is involved in the *p21* suppression, resulting in the enhancement of p53-mediated autophagy and apoptosis [58]. Moreover, *MDM2* also is suppressed by adenoviral E1A accumulation through an unknown mechanism [24].

## 8. Clinical Relevance and Future Perspectives

A Phase I clinical trial of OBP-301 was conducted to determine the safety, local response, and pharmacodynamics of OBP-301 as a monotherapy in patients with advanced solid tumors, including leiomyosarcoma and neck sarcoma, in the United States [101]. Intratumoral injection of OBP-301 was well tolerated by patients with advanced solid tumors, as there were no severe adverse events. Although the neutralizing antibody titer was increased after OBP-301 treatment in all patients, one patient had a partial response and seven patients had stable disease. These findings demonstrate that intratumoral injection of OBP-301 is a useful method in cancer patients with neutralizing antibodies to adenovirus. As OBP-301 exhibits a chemosensitizing effect in sarcoma cells [54], multidisciplinary therapy with OBP-301 and chemotherapy is expected to be a promising antitumor strategy for the treatment of bone and soft-tissue sarcomas.

Metastatic tumors are a main obstacle to improving the clinical outcome of patients with bone and soft-tissue sarcomas. Recently, immune checkpoint inhibitors (ICIs) have improved the clinical outcomes of certain cancer patients with metastatic tumors; however, patients with bone and soft-tissue sarcomas are less sensitive to ICI therapy [102]. The refractoriness of bone and soft-tissue sarcomas to ICI therapy may be due to the characteristics of non-inflamed “cold” tumors with low mutation burdens [103] and low mismatch-repair deficiencies [104]. Oncolytic virotherapy has recently emerged as an immune modulator for enhancing the antitumor immune response [105,106,107]. Oncolytic adenoviruses induce immunogenic cell death through the release of damage-associated molecular patterns, such as adenosine triphosphate, high-mobility-group B1 proteins, and uric acid [108]. As metastatic osteosarcoma tumors have been shown to exhibit more abundant mutation burdens than primary tumors [109], combination therapy with oncolytic adenoviruses and ICI may become a novel option for the treatment of metastatic osteosarcomas. Further preclinical experiments would be warranted to evaluate the therapeutic potential of combination immunotherapy with oncolytic adenoviruses and ICI for the treatment of metastatic osteosarcomas.

## 9. Conclusions

Ad5 is a useful tool for generating genetically bioengineered tumor-specific replicative oncolytic viruses. Based on the structure of the Ad5 genome, we have generated three types of *hTERT* promoter-driven replication-competent oncolytic adenoviruses, including OBP-301, the GFP-expressing OBP-401, and the *p53*-armed OBP-702. OBP-301 is a strong inducer of autophagy-related cell death in telomerase-positive tumor cells. Further exploration of the precise mechanism of OBP-301-induced autophagy would facilitate improvements in the therapeutic potential of OBP-301-based oncolytic virotherapy in bone and soft-tissue sarcomas. GFP-expressing OBP-401 is a useful tool for the assessment of CAR-positive tumor cells that are suitable targets for OBP-301-based oncolytic virotherapy. OBP-401-based FGS would be a novel therapeutic option for the treatment of bone and soft-tissue sarcomas. *p53*-armed OBP-702 is a next-generation oncolytic virus that is expected to overcome the refractoriness of tumors to OBP-301 and to suppress tumor-related bone destruction in osteosarcoma. OBP-702-based oncolytic virotherapy would be a promising antitumor strategy for the treatment of osteosarcoma with an invasive phenotype. The clinical application of telomerase-specific oncolytic virotherapy is ongoing for the treatment of cancer patients. Thus, *hTERT* promoter-driven oncolytic adenoviruses are expected to provide novel therapeutic options for the treatment of bone and soft-tissue sarcomas.

## Figures and Tables

**Figure 1 cancers-12-00478-f001:**
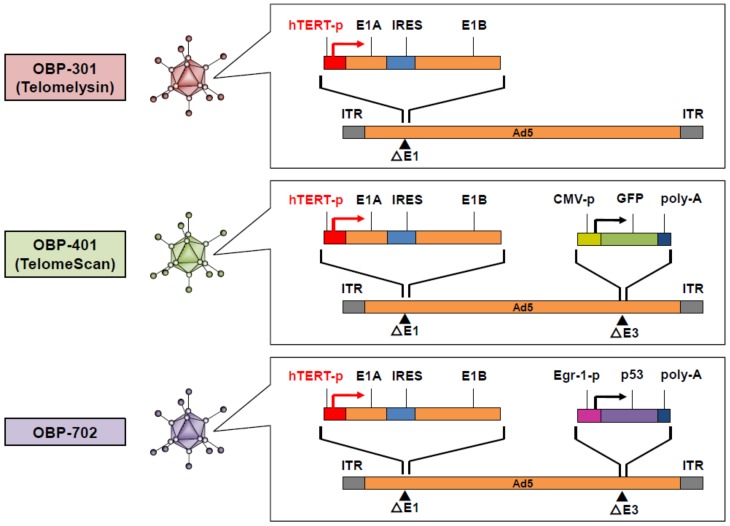
Structures of *hTERT* promoter-driven oncolytic adenoviruses. hTERT: human telomerase reverse transcriptase; IRES: internal ribosome entry site; GFP: green fluorescent protein; ITR: inverted terminal repeat.

**Figure 2 cancers-12-00478-f002:**
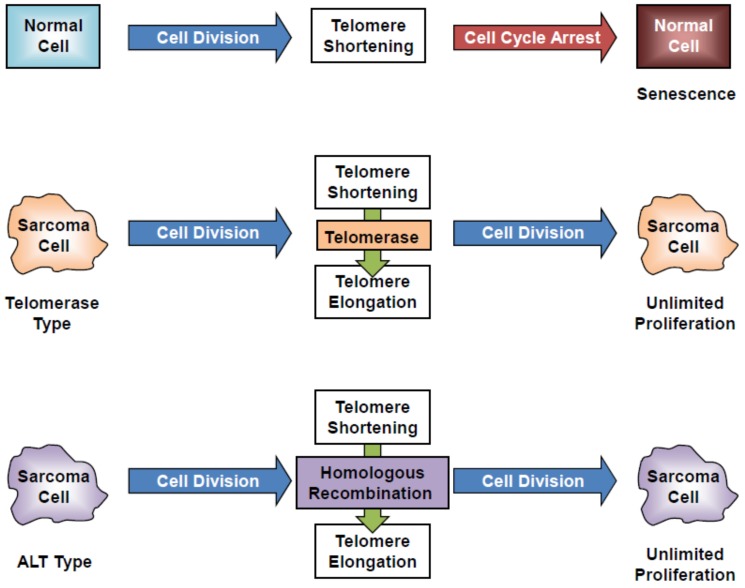
Telomerase-dependent and telomerase-independent telomere maintenance in sarcoma cells. ALT: alternative lengthening of telomere.

**Figure 3 cancers-12-00478-f003:**
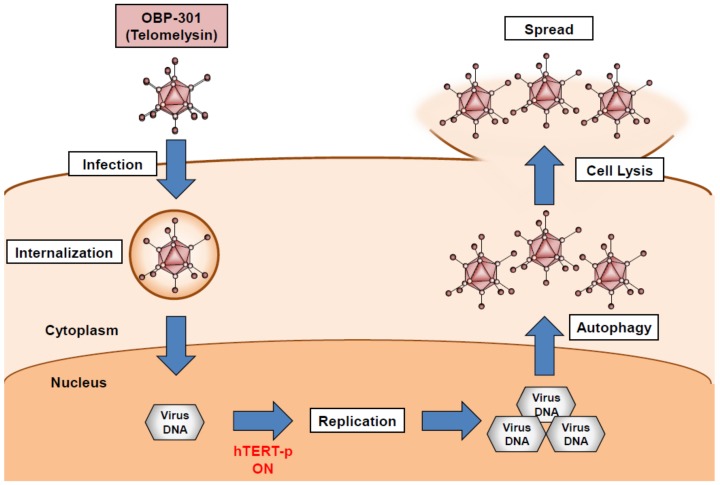
OBP-301-mediated cell death process in association with autophagy. hTERT: human telomerase reverse transcriptase.

**Figure 4 cancers-12-00478-f004:**
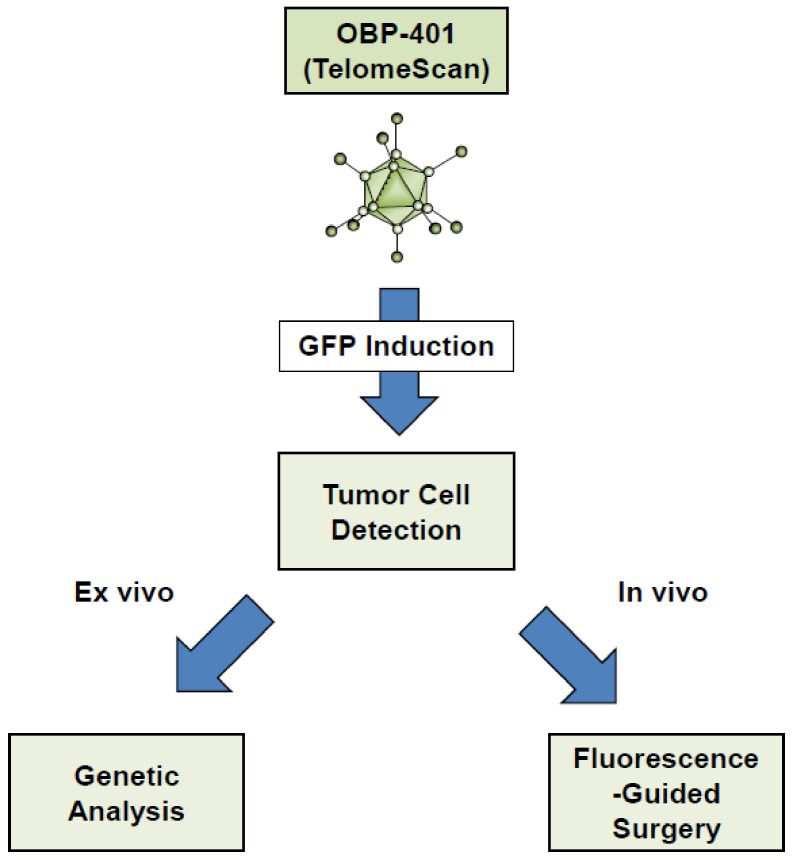
OBP-401-mediated tumor-specific imaging system.

**Figure 5 cancers-12-00478-f005:**
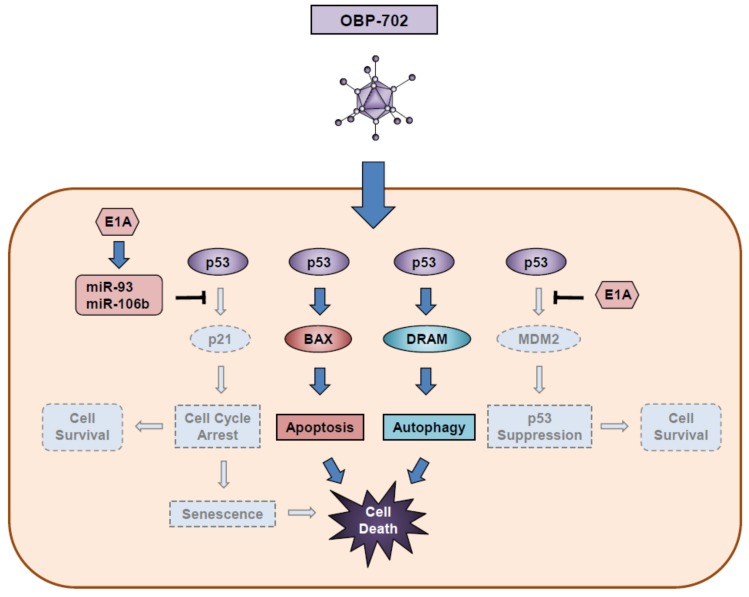
OBP-702-mediated cell death process in association with activation of p53 signaling pathway. BAX: BCL2 associated X apoptosis regulator; DRAM: damage-regulated autophagy modulator.

**Table 1 cancers-12-00478-t001:** Characterization of human bone and soft-tissue sarcoma cell lines.

No.	Cell Line	Tumor Origin	Tumor Type	hTERT Expression *	CAR Expression **	OBP-301 Sensitivity ***
1	OST	Bone sarcoma	Osteosarcoma	++	+++	+++
2	U2OS	Bone sarcoma	Osteosarcoma	+	+++	++
3	HOS	Bone sarcoma	Osteosarcoma	++	+++	++
4	HuO9	Bone sarcoma	Osteosarcoma	+	++	++
5	MNNG/HOS	Bone sarcoma	Osteosarcoma	++	+	+
6	SaOS-2	Bone sarcoma	Osteosarcoma	+	+++	+
7	NOS-2	Bone sarcoma	Osteosarcoma	++	+++	++
8	NOS-10	Bone sarcoma	Osteosarcoma	++	+	++
9	NDCS-1	Bone sarcoma	Chondrosarcoma	+++	++	++
10	OUMS27	Bone sarcoma	Chondrosarcoma	+	ND	None
11	CCS	Soft-tissue sarcoma	Clear cell sarcoma	+++	++	+++
12	NMS-2	Soft-tissue sarcoma	Malignant peripheral nerve sheath tumor	++	++	++
13	SYO-1	Soft-tissue sarcoma	Synovial sarcoma	+++	+	++
14	NMFH-1	Soft-tissue sarcoma	Malignant fibrous histiocytoma	+	ND	None

* Relative hTERT mRNA expression: + = < 0.1, ++ = 0.1-10, +++ = 10 <. ** Mean fluorescence intensity of CAR: ND = not detectable, + = < 100, ++ = 100-200, +++ = 200 <. 4. Antitumor Effect of OBP-301 in Association with Autophagy. *** ID_50_ values of OBP-301 on day 5 after infection: + = 60 Multiplicity of infection (MOI) <, ++ = 20-60 MOI, +++ = < 20 MOI.

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
