# Peer review of "Bone and Soft-Tissue Sarcoma: A New Target for Telomerase-Specific Oncolytic Virotherapy"

_cancers, 2020, doi:10.3390/cancers12020478_

Round 1

Reviewer 1 Report

The review article by Hiroshi Tazawa et al. described the telomerase-specific oncolytic virus therapy for bone and soft-tissue sarcoma, which presented interesting aspects to understand the recent progress in the fields. However, there are several minor comments that need their attention to address. After the minor revision/correction, the manuscript will be further strengthened to be suitable enough for the publication in the Cancers with a benefit to the readers. My minor comments are enclosed below.

Minor comments.

1.      This review article is focusing on Sarcoma therapy but there is no description about Sarcoma in the introduction section. In the introduction, please describe about sarcoma and why TERT-targeting oncolytic adenovirus has benefit for sarcoma.

2.      Similarly, this review article is focusing on Telomerase-specific oncolytic virus therapy for Sarcoma, but, there is lack of description for reason of TERT-targeting oncolytic virus for Sarcoma. For example, TERT activation and mutation status...etc.  Please describe details.

Reviewer 2 Report

This is a very interesting review article describing how telomerase-specific oncolytic virotherapy may serve patients diagnostically and therapeutically.  There are several comments and questions to address:

It would be helpful if the authors would be more specific with which bone and soft tissue sarcomas are referenced throughout the paper rather than simply stating “sarcomas”.  For example, under “CAR Expression in Bone and Soft-tissue Sarcoma cells”, the reader would like to know which bone and soft-tissue sarcomas are being referenced.  With regards to the cell lines, the reader would like to know what the cell lines listed represent.  For example, what kind of cell line is NMS-2?  Furthermore, under “Fluorescence-Guided Surgery using the OBP-401-based GFP Induction System”, the authors ought to elaborate on osteosarcoma and soft-tissue sarcoma rather than mentioning a variety of solid tumors and focusing on pancreatic cancer.  What findings are there in what specific bone and soft-tissue sarcomas?  The authors seem to focus on osteosarcoma occasionally throughout the paper but otherwise are very broad.

The authors point out it would be critical to assess CAR expression in tumor cells before initiating treatment; however, what about other potential pitfalls of treatment with Ad5 exist?  For example, Zhao H, et al., 2018, reported neutralizing antibodies to human Ad5 in blood samples obtained from patients with cancer.  What about side effects?

For the section on “Antitumor Effect of OBP-301 in Association with Autophagy”, the authors go into detail about the underlying mechanisms behind adenovirus Delta-24-RGD but make no actual mention of mechanisms for OBP-301.  What studies have been down to examine these mechanisms/downstream targets for OBP-301?

Under “Combination Therapy with OBP-301 and Chemotherapeutic Reagents for Osteosarcoma”, the citations used for the third sentence regarding gemcitabine and paclitaxel are for lung and gastric cancers, not for osteosarcoma.  The jump to application in osteosarcoma is a large one.  The authors may consider citing Conry R, et al., 2016, regarding the use of zoledronate specifically in patients with metastatic osteosarcoma.

The section on p53 signaling would benefit from additional citations on the role of p53 mutations and osteosarcoma. 

In the section on clinical relevance, the authors make another large leap by extrapolating radiosensitization findings in patients with esophageal cancer to patients with bone and soft-tissue sarcomas.  What sarcomas were examined in the Phase I clinical trial?  Are there are clinical studies regarding radiation, OBP-301, and bone and soft-tissue sarcomas?  The authors also need citations regarding the possible role of ICIs in combination with oncolytic adenoviruses in patients with bone and soft-tissue sarcomas.

Round 2

Reviewer 2 Report

A few minor revisions and this paper would then be ready for publication:

Remove “(except for OUMS-27 and NMFH-1)” from line 122 as this only adds confusion.

Remove “or zoledronic acid (ZOL [55]” from line 163 and then replace “ZOL” with “Zoledronic acid (ZOL)” in line 173.

Revise line 211 to say “solid tumors [79, 85-89], including osteosarcoma [90] and soft-tissue sarcoma [91].”

Remove “, suggesting the antitumor effect of OBP-401 in residual sarcoma cells.  A patient-derived orthotopic xenograft (PDOX) model of pancreatic cancer has shown that 216 PDOX tumors can be visualized by the OBP-401-based illumination system [92]”. 

Replace “In contrast” with “Additionally” for line 236. 
